# Application of Nitrogen Piezoelectric Direct Discharge for Increase in Surface Free Energy of Polymers

Dariusz Korzec *[ID], Florian Hoppenthaler, Thomas Andres, Sophia Guentner [ID] and Simona Lerach

Relyon Plasma GmbH, Osterhofener Straße 6, 93055 Regensburg, Germany;
f.hoppenthaler@relyon-plasma.com (F.H.); t.andres@relyon-plasma.com (T.A.);
s.guentner@relyon-plasma.com (S.G.); s.lerach@relyon-plasma.com (S.L.)
* Correspondence: d.korzec@relyon-plasma.com

**Abstract:** The subject of this study is the application of the piezoelectric direct discharge (PDD) operated with nitrogen to control the surface free energy (SFE) of polymers. The activation area, defined as the area of the zone reaching the SFE of 58 mN/m for high-density polyethylene (HDPE) and poly (methyl methacrylate) (PMMA), is characterized. For HDPE, the activation area was characterized as a function of the distance from 1 to 16 mm, the nitrogen flow from 5 to 20 SLM, and the treatment time from 1 to 32 s. For larger distances, where SFE does not exceed 58 mN/m, the water contact angle is evaluated. The activation area for nitrogen PDD is typically a factor of 3 higher than for air with all other conditions the same. A maximum static activation area of 15 cm$^2$ is reached. The plasma treatment of lens panels made of PMMA is presented as application example.

**Keywords:** atmospheric pressure plasma (APP); resonant piezoelectric transformer; piezoelectric direct discharge (PDD); high-density polyethylene (HDPE); surface free energy (SFE)

## 1. Introduction

Cold atmospheric pressure plasma jets (APPJ) are very important and widely used tools for surface treatment [1–7]. Typically, the APPJs operated with air produce plasma which is small in size [8–11]. Their activation area can be increased using the noble gases, especially He [12–18] and Ar [19–23], or nitrogen [24–29].

Application of nitrogen ionization gas is an attractive alternative to noble gases because it is not only more economical, but also has excellent process properties. It is widely used for increasing the hydrophilicity of hydrophobic surfaces [30]. Nitrogen plasma can be used for hydrophilization of polyethylene [31]. For different kind of plasmas, a contact angle much lower than for air or oxygen plasma is demonstrated [31,32]. A strong increase in the polar component of the surface free energy (SFE) is observed when nitrogen is applied instead of CDA for pulsed arc APPJ treatment [33]. The SFE increase correlates with improved process performance, e.g., an increase in adhesion of pressure-sensitive adhesive (PSA) on polymers [33], very fast nitrogen APPJ treatment of composite layers [24], or rapid annealing (oxidation) of SnO$_x$ layers [34].

The superior properties of nitrogen discharge are based on different mechanisms than for noble gases. Both Ar and He have much lower breakdown voltage than air, meaning that plasma can be generated at a much lower voltage and can reach larger sizes at atmospheric pressure. This mechanism does not work for nitrogen, because it has a higher breakdown voltage than air [35].

Essential for activation of polymer surfaces is the presence of small amounts of oxygen in the plasma allowing for generation of reactive oxygen and nitrogen species (RONS). In air discharges, the RONS are produced directly in the primary plasma. In nitrogen discharges, the indirect mechanisms play an important role.

In many types of cold atmospheric pressure nitrogen plasmas, the role of the metastable excited state $N_2(A^3\Sigma_u^+)$ is revealed as the reason for wide spreading of the discharge. The atmospheric pressure DBD in nitrogen is based on Townsend discharge, in which metastable molecules stay in the gas and create electrons through cathode secondary emission. A significant contribution of the Penning ionization is indicated [36]. In $N_2$-pulsed positive corona discharge, $N_2(A^3\Sigma_u^+)$ metastable molecules are mostly generated in the primary streamer channels [37]. The APP cold nitrogen discharge can exist in both filamentary and glow mode [38]. Significant for the energy transfer in the atmospheric pressure nitrogen plasma afterglow is the presence of a much higher vibrational temperature than air [39].

One of the cold atmospheric pressure plasmas recently gaining popularity due to its efficiency, compactness and versatility is the piezoelectric direct discharge (PDD) [40,41]. It was characterized for operation of APPJ with ambient air [42,43], and is widely used for surface treatment. The few examples of PDD applications are the sintering for inkjet-printed metallic traces in 3D printed electronics [44], the surface modification of the adhesive bonding of polyolefins [45], treatment for improved adhesion of orthopedic materials [46], improvement in wetting the $SiO_2$[47] surface on silicon wafers, promoting the osseointegration and biodecontamination of nanostructured titanium implants [48,49] and PEEK implants [50], treatment of medical implant-grade PEEK composites [51], in vitro tests on the dentin of human teeth [52], sterilization of 3D objects [53], aerosol charging [54], or generation of negative ions [55].

It is known that humidity has a strong influence on the chemistry of the APPJ [56]. To avoid the undefined influence of humidity in ambient air on the process result, the plasma device piezobrush® PZ3-i was developed, allowing for operation with industrial gases such as compressed dried air (CDA) or nitrogen. The aim of this work is the characterization of this instrument for surface activation of polymers with nitrogen PDD and comparison with results for CDA. Plane substrates made of high-density polyethylene (HDPE) are used as a model material for a systematic parameter study conducted using the activation image recording (AIR) method [57] and water contact angle measurement [58]. The nitrogen PDD treatment of poly (methyl methacrylate) (PMMA) lens modules is used as a practical application example.

## 2. Materials and Methods

### 2.1. Plasma Treatment

In this study, the prototype of the commercial device piezobrush® PZ3-i of Relyon Plasma GmbH was used. The core component of this device is the piezoelectric cold plasma generator (PCPG) of the type CeraPlas™ F produced by TDK Electronics GmbH, Austria. It is a resonant piezoelectric transformer with maximum input power of 8.0 W operating at a resonance frequency (second harmonics) of 50 kHz, and is used for high voltage generation. The surface activation is performed by piezoelectric direct discharge (PDD), described in detail in our previous work [43]. In opposite to the handheld device piezobrush® PZ3 [43,59], the piezobrush® PZ3-i is designed for operation with industrial gases such as CDA or nitrogen instead of ambient air.

Figure 1a shows a schematic setup for substrate surface activation, with the distance *d* between the treated surface and the tip of the PCPG. The plasma liner shown in the picture included in Figure 1a is made of polybutylene terephthalate (PBT). The tubular plasma liner extension used for water contact angle measurements in Section 3.4 is made of polyethylene terephthalate (PET) foil. The PCPG power is switched on for a short, predefined time, typically 10 s, however, the nitrogen flow is established for 1 min before the first switching on of the power.

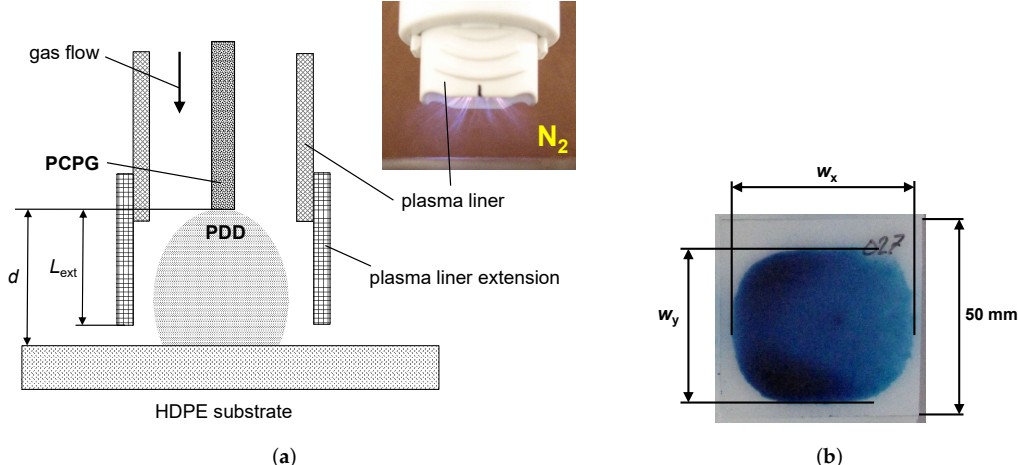

**Figure 1.** Surface activation. (**a**) Setup for activation of the polymer substrate surface, where the distance $d$ means the distance between the substrate and the tip of the PCPG, $L_{ext}$ is the effective length of the plasma liner extension, and (**b**) is the visualization of a typical activation area with SFE $\geq$ 58 mN/m produced by 10 s nitrogen PDD at a distance $d = 3.5$ mm, nitrogen flow of 8 SLM and PCPG input power of 8.0 W.

### 2.2. Activation Area Determination

Figure 1b shows the visualization of the zone activated on the surface of the HDPE substrate using the setup shown in Figure 1a. The 58 mN/m test ink is chosen to ensure that the ink is only wetting the activated surfaces and rolls off the nontreated surface areas. The width of the activation zone $w_x$ (see the symbols in Figure 1b) is larger than the width $w_y$, following the elongated shape of the PDD caused by the rectangular geometry of the PCPG tip.

The area of the activation zone visualized by the test ink is used for quantitative evaluation of the plasma source performance. About 16 μL of test ink is distributed evenly across the activation zone by use of a small brush. Since the amount of ink affects the measured activation area [57], it is important to keep the the ink distribution process reproducible. The automated method of ink patch area determination, the activation image recording (AIR), described in detail elsewhere [57], is used. Since the area of the ink patches changes over time, the pictures of the ink patch are taken in short intervals by the use of a digital camera. The contour of the ink patch is automatically recognized, and the number of pixels is counted. By comparison with the number of pixels of the known area of the entire substrate, the actual area of the test ink patch is calculated.

### 2.3. Water Contact Angle

The AIR method is very efficient for treatment conditions assuring the surface activation with SFE larger than 58 mN/m. For weak activation, e.g., conducted in a large distance, when SFE is below 58 mN/m, the AIR method is not suitable and an alternative evaluation method should be used. In this study, the complementary method is the determination of water contact angle.

However, the contact angle cannot be considered as a replacement for AIR. On the one hand, it does not provide any information about the activated area. On the other hand, it is difficult to apply the water contact angle method for strong activation, because very small differences between contact angles for different parameters are difficult to evaluate. This is why in this study contact angles are only measured for the cases when AIR does not perform well.

Double-distilled water is used to produce the test droplets of about 4 μL on the substrate surface. The droplets are dispensed at the surface position corresponding to the symmetry axis of the PCPG during treatment. The side-view macroshots are made with a Canon G10 digital camera, and the contact angles are determined graphically, as shown in the examples presented in Figure 2.

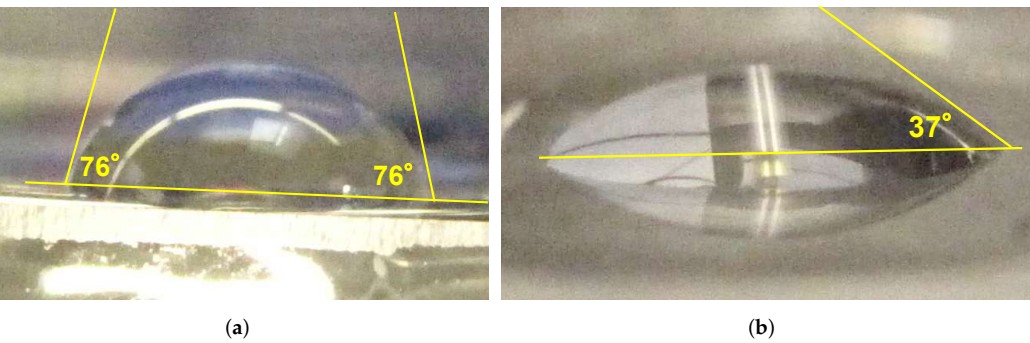

(a)　　　　　　　　　　　　　　　　　　(b)

**Figure 2.** The droplets of the distilled water applied on the flat side of the PMMA lens (**a**) before treatment, and (**b**) after plasma treatment with nitrogen PDD at a distance of 6 mm, nitrogen flow of 8.0 SLM, and treatment time of 3 s.

*2.4. Test Inks*

Different liquid mixtures are used for production of test inks suitable for determination of the SFE of solid surfaces. The test inks defined in a number of standards [60] and gauging the surface energies from 31 to 56 mN/m are mixed of formamide and 2-ethoxyethanol. For gauging the test inks in the SFE in the range from 60 to 72 mN/m, the mixtures of formamide with DI water are used. In this study, such formamide-based test inks are provided by ArcoTest GmbH. The 58 mN/m test ink is based on pure formamide. The reason for choosing this ink as a standard for the AIR method and its limitations (ageing, environmental influences, shrinkage curves) are discussed in detail in [57].

*2.5. Substrates*

The substrates used for systematic investigation of the PDD performance are made of "natural" HDPE with sizes 100 mm × 50 mm × 2 mm or 50 mm × 50 mm × 2 mm supplied by Rocholl GmbH, Germany. Following the standard [60], the substrates are preconditioned after delivery for at least 40 h under 23 °C and 50% humidity.

Different preparation procedures for the HDPE surface before plasma treatment are known. For example, they are cleaned by ultrasonic rinsing in iso-propanol for 30 s [61], in distilled water for 20 min [62], or in 96% ethanol for 5 min and dried in dynamic vacuum at 0.1 Pa for 10 min [63], or purified by extraction with acetone overnight and stored in a desiccator prior to each plasma treatment [64]. To avoid the influence of solvents and water on the result, no wet pretreatment of the substrates was applied.

For the HDPE substrates cleaned dry by wiping with paper tissue, to remove the dust particles and residues of sawdust, the visualized activation area is on average 5% larger than for the pristine ones. This source of error can be avoided by using the pristine substrates.

In the application example, the panels of lenses made of PMMA are used.

All substrates are treated with plasma and exposed to the test ink at a temperature 23 °C ± 2 K and relative humidity of 50% ± 5%.

*2.6. Activation Area*

The aim of the polymeric surface activation is to increase the number of functional groups. The scenarios of activation or functionalization of HDPE by different types of plasma have some common features. First, the electron-impact dissociation of surface hydrogen atoms creates dangling bonds prone to docking of the functional groups. In

the second step, depending on the concentration of different species, the saturation of these bonds with functional groups occurs. The FTIR measurements show in the HDPE treated by oxygen containing plasma the presence of functional groups such as C–O, C=O, O–C–O [64] or –OH [65]. The XPS measurements show strong increases in the spectral peaks of C–O, C=O, and O–C=O [66]. To increase the SFE of HDPE, both the increase in the concentration and mean energy (temperature) of electrons and an abundance of oxidizing chemical radicals are needed.

PDD Morphology

The activation area correlates with the expansion of the plasma plume generated by an APPJ. The zone of the strong illumination correlates with the primary plasma, where the intensive ionization, electronic excitation and relaxation occur. The primary plasma is also where the chemically active species and metastable molecules are produced. The pictures of discharge plumes provide information on where the most efficient surface activation can be expected. In Figure 3, the PDD plumes generated with CDA and with nitrogen are compared. The nitrogen PDD plume expands more in both directions: depth and width, reaching 14 mm and 39 mm, respectively, which is 36% and 26% larger than for CDA, respectively. These percentages are underestimated, because the CDA discharge is much darker than the nitrogen one and a much longer (1 s) exposure time than for nitrogen (0.3 s) was needed to obtain the picture. A much larger expansion of the nitrogen PDD implies that a much larger activation area after treatment with nitrogen PDD than for CDA can be expected.

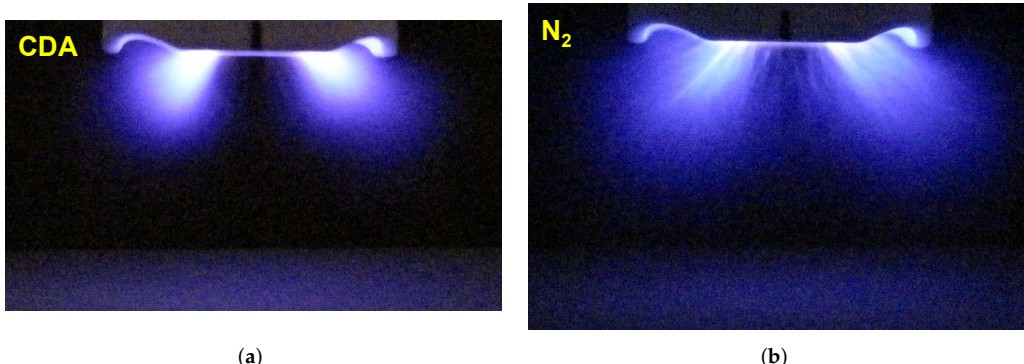

(a)          (b)

**Figure 3.** The PDD in (**a**) CDA, the camera exposure time was 1 s, and (**b**) in nitrogen, the camera exposure time was 0.3 s, for PCPG power of 8.0 W, gas flow of 8 SLM, and distance of 20 mm. The lens aperture for both pictures was 2.8.

### 3. Results and Discussion

*3.1. SFE and Activation Area*

The chemically active species causing the surface activation are not distributed homogeneously across the plasma plume of the APPJ. Therefore, the SFE varies across the activated zone. Applying the test inks gauged with different SFE, ink patches of different size are obtained. Figure 4 shows the visualized activation area obtained by PDD treatment of HDPE for air and nitrogen as a function of the SFE gauged by the test inks. Considering the fitting lines, the size of the visualized activation area is is more than double when the 46 mN/m test ink is taken instead of the pure water-based 72 mN/m test ink. The visualized area for 46 mN/m is 37% larger than for 58 mN/m. In the investigated FSE range, the activation area for nitrogen is a factor 3.3 for 46 mN/m to 4.1 for 72 mN/m larger than for CDA. For different applications, different minimum SFE levels are required and used for definition of the activation area.

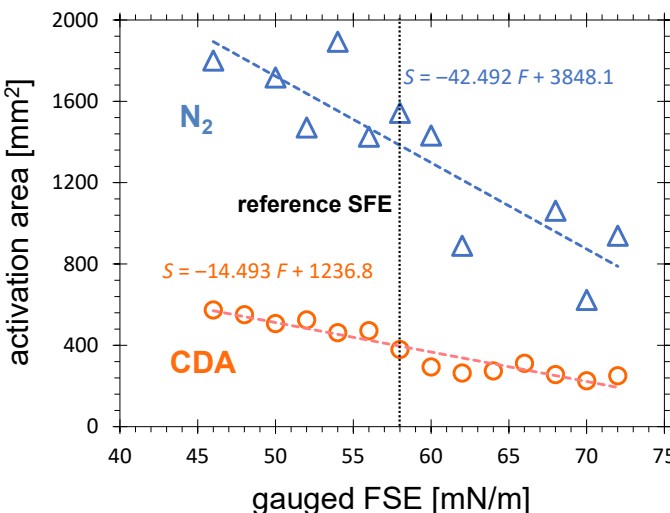

**Figure 4.** The activation area on HDPE substrates treated for 10 s by CDA or nitrogen PDD at a distance of 5.5 mm from the PCPD tip, gas flow of 8.0 SLM and visualized with test inks gauged with SFE from 46 to 72 mN/m.

### 3.1.1. Influence of Distance

Figure 5 shows the dependence of the activation area on the distance between the tip of the PCPG and the HDPE substrate for PDD operated with nitrogen and CDA. The common feature of both curves is that the activation area monotonously decreases with increasing distance. Since the surface activation is caused mainly by oxidizing species such as atomic oxygen, nitrous oxides, and OH groups, from the presented results it can be concluded that the concentration of these species decreases with distance from the primary discharge. In general, the reasons for this are as follows:

(i) The electric field weakens. The electric field produced around the tip of the PCPG gets weaker approximately inversely proportionally to the square of the distance. Consequently, the energy transfer from the electric field to the electrons diminishes and the production rates of the chemically active species decrease. Since the electric field produced by PCPG is very strong [55], this mechanism plays an important role in our case.

(ii) The decay of short-living chemically active species. Most of the oxidizing radicals produced in the primary discharge have a lifetime which is comparable with the time of transfer from the primary discharge to the substrate surface. The decay of the short-living species is exponential with the time they stay in the gas. Consequently, the larger the distance from the primary discharge, the lower their concentration.

(iii) The decay of excited species. The most of electronically excited species are so short living that they are already relaxed in the bulk of the primary discharge. However, some metastable excited species have lifetimes allowing them to transfer the energy from the primary discharge to the more remote regions, thus they can contribute to Penning-type ionization or dissociation. Furthermore, their decay is exponential with distance.

(iv) The geometrical effects. The species produced in the primary discharge spread in space, resulting in a dilution of the active species in increasing volume. This mechanism is valid not only for short-living but also for considerably stable oxidizing species, such as $O_3$, $NO_2$ or $H_2O_2$, which have lifetimes much longer than the transfer time from the primary discharge to the substrate (e.g., ozone—many hours). The rapid decrease in the $O_3$ and $NO_2$ concentrations with distance from the PCPD operated in air is documented in Figure 16 in [42]. These results show that the concentration of ozone is in general a factor of about 20 higher than nitrogen dioxide.

(v) The thermal effects. It is known that the treatment temperature plays an important role in the HDPE surface treatment. When the treatment temperature is close to the melting point of the polymer, surface molecular motion is not negligible [67] and promotes chemical reactions. Additionally, the reactivity of the oxidizing species at the polymer surfaces increases exponentially with the temperature. The gas temperature in the primary discharge is higher than the ambient one, and decreases with increasing distance due to mixing of the warm plasma gas with the colder surrounding gas. Consequently, the amount of oxidizing events at the substrate surface will decrease with the distance. Since the temperature of the gas coming out of the PDD discharge is only a few tens of K higher than the ambient temperature, this mechanism is not dominant in our case. Furthermore, the thermal load of the substrate is very similar in CDA and nitrogen, and can not explain the difference between the CDA and $N_2$ curves in Figure 5.

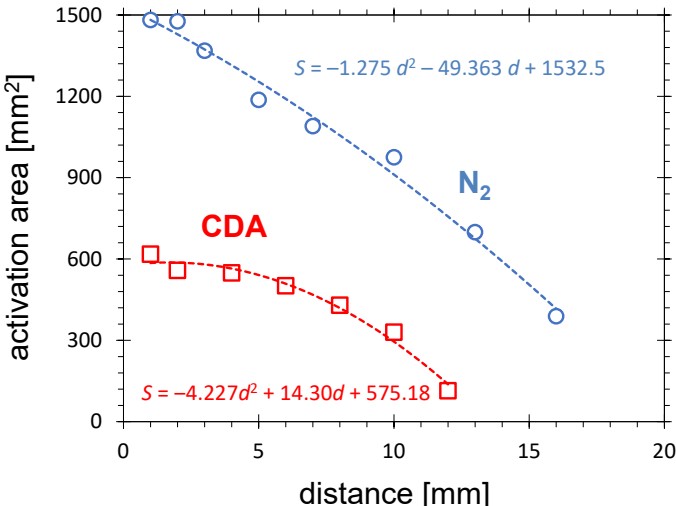

**Figure 5.** The activation area after $N_2$ or CDA PDD treatment as a function of distance between the PCPG tip and the HDPE substrate for the gas flow of 8.0 SLM, the treatment time of 10 s, and the PCPG input power of 8.0 W.

Using the fitting formulas from Figure 5, the distance at which the activation area vanishes can be estimated. It is 13.5 mm and 20.4 mm for CDA and nitrogen, respectively. These values are slightly larger than the sizes of the primary discharge estimated on the basis of the PDD pictures in Figure 3.

The comparison of both curves in Figure 5 makes it apparent that the activation area produced by nitrogen PDD is much larger than for CDA PDD. At the distance of 1 mm, the activation area for nitrogen is on the factor of 2.5 (1500 mm$^2$) larger than for CDA (600 mm$^2$). This result seems surprising. Since in CDA the concentration of oxygen is much higher, the higher concentration of oxidizing radicals can also be expected. Considering the important oxidizing species, ozone, its concentration increases with the oxygen percentage in the gas mixture with nitrogen. In pure (without any contact to the ambient) nitrogen discharge, no ozone can be measured (see Figure 6b in [41]). However, in the case of operation of APPJ in ambient air, the excited nitrogen molecules and other energy-carrying species can cause an efficient production of the oxidizing radicals at a small distance from the substrate surface.

Three mechanisms contribute to the larger nitrogen plasma plume.

The first mechanism results from a higher concentration of the long-living, metastable, excited molecules $N_2\left(A^3\Sigma_u^+\right)$ in the nitrogen plasma, allowing the transfer of energy with the gas flow at a larger distance. These species contribute to the Penning ionization at a larger distance from the primary plasma and to the generation of RONS. The addition of a very small amount of $O_2$ (less than 175 ppm) in $N_2$ DBD discharge is sufficient to completely change the surface chemistry: N is replaced by O, and the level of transformation significantly decreases [68].

The second mechanism is based on the strong electric field present in a distance up to 30 mm from the PCPG tip [55]. This causes the PDD extension because the presence of high-energetic excited molecules promote the ionization processes. The oxygen molecules cause the quenching of the metastable excited nitrogen. Consequently, the field effect will be more pronounced in nitrogen with small admixture of oxygen.

The third mechanism, valid only in small distances from the PCPG tip ($<$10 mm) is the photoinduced dissociation [69].

### 3.1.2. Influence of Gas Flow

The dependence of the activation area on the nitrogen flow is quite different compared with the CDA curve. Figure 6 shows that the surface area doubles when the nitrogen flow increases from 5 to 20 SLM. In the same flow range, a slight decrease in the activation area for CDA is observed.

The shape of the CDA curve is determined by two competing processes. On the one hand, at a given production rate for long-living, chemically active species at the PCPG tip, with increasing flow their concentration decreases due to dilution. As an example, the ozone concentration in gaseous plasma products of the PDD is almost inversely proportional to the flow [43]. On the other hand, with increasing gas flow, the transfer time of chemically active species from the primary PDD to the substrate decreases, giving the short-living, chemically active species (reactive oxygen and nitrogen species—RONS) the chance to reach the substrate and perform the surface activation before they recombine in the gas volume.

It is also important to note that for the distance of 5.5 mm between the PCPG tip and the substrate, for which the curves in Figure 6 are measured, a strong electric field is still present at the surface of the substrate. Since this field is not gas-flow dependent, the activation area related to the primary plasma does not decrease either, due to dilution of the chemically active species. These species are generated effectively in the direct vicinity of the substrate.

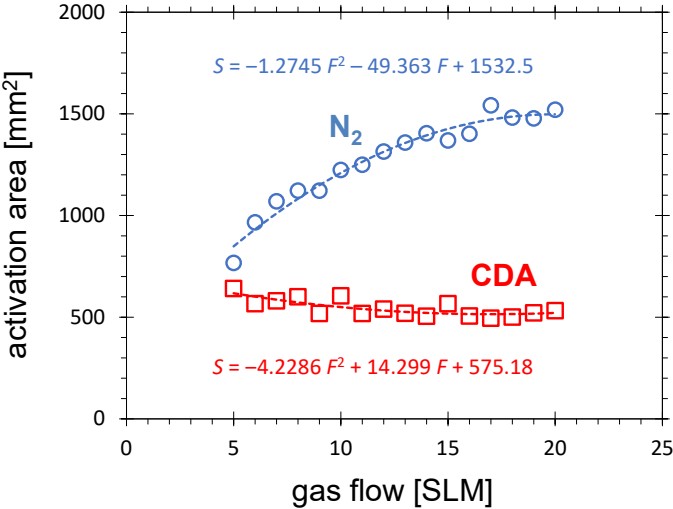

**Figure 6.** The activation area after $N_2$ or CDA PDD treatment as a function of gas flow for the distance between the PCPG tip and the HDPE substrate of 5.5 mm, the treatment time of 10 s, and the PCPG input power of 8.0 W.

By operation of the PDD with nitrogen, an additional mechanism should be considered. The amount of metastable nitrogen decreases rapidly with increasing oxygen concentration because oxygen is an electronegative gas which has a very strong electron affinity [69]. Since the main loss mechanism of nitrogen metastable excited molecules is quenching with oxygen from the ambient air and humidity, with increasing nitrogen flow, the influence of this loss mechanism diminishes. The excited nitrogen species can overcome a larger distance before giving their energy up to chemistry-generating processes. By operation

of the PDD with CDA, this mechanism is not active, because the concentration of oxygen does not change with increasing gas flow.

### 3.1.3. Influence of Treatment Time

Figure 7 shows the time dependence of activation area on treatment time for CDA and nitrogen respectively. Both curves show a monotonous increase with treatment time and saturation for longer treatment. This limitation is observed for different types of APPJ [57,70] and can be explained by three mechanisms: (i) increasing dilution of the chemically active species in the ambient air with increasing distance from the nozzle opening blowing the plasma gases, (ii) the decrease in the concentration of the short-living chemically active species with flow time due to recombination and quenching processes, and (iii) the decrease in the CeraPlas™ F electric field with increasing distance from the PCPG tip.

The activation area produced by nitrogen PDD is a factor of about 3 larger than for air. The reason for this discrepancy was already discussed in Section 3.1.1. This factor decreases with increasing treatment time, starting with 3.8 for 1 s and reaching 2.8 for 10 s of treatment. This decrease can be explained either geometrically or thermally. The relative area increase through the same linear increase is much larger for smaller activation areas of CDA than for large areas of nitrogen. The nitrogen has lower thermal conductivity than air. This results in degraded cooling of the PCPG and consequently less efficient discharge [59].

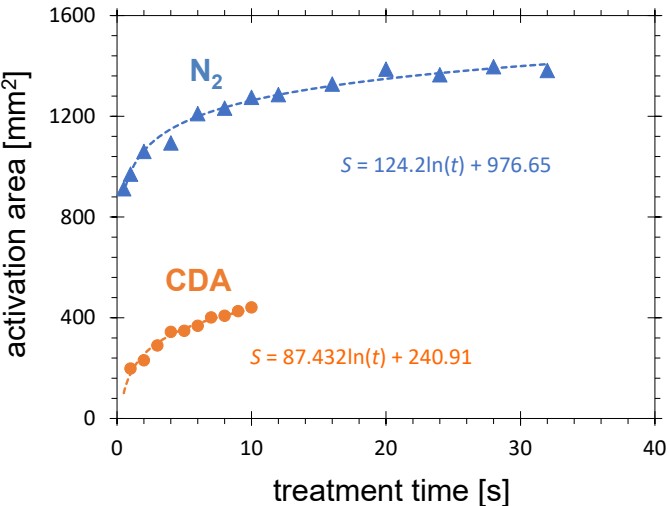

**Figure 7.** Influence of treatment time on the activation area and for CDA and $N_2$ flow of 8.0 SLM, distance between the PCPG tip and the HDPE substrate of 5.5 mm, and the PCPG input power of 8.0 W.

### 3.2. Activation Rate

The activation rate $A_{rate}$ can be defined [71] as the ratio of the activated area $S_{act}$ to the treatment time $t_{treat}$:

$$A_{rate} = \frac{S_{act}}{t_{treat}} \qquad (1)$$

A planar substrate can be plasma-treated with APPJ dynamically or statically. The dynamic treatment means that the APPJ moves relative to the substrate during the plasma-ON time. The dynamic treatment can be realized either by fixing the APPJ and moving the substrate, e.g., with the use of a belt conveyer, or by fixing the substrate and moving the APPJ, e.g., by hand or by robot. Due to the changes in the speed and an uncontrollable air movement, it is difficult to create reproducible conditions for measurement of activation rate for dynamic plasma treatment.

Much better reproducibility for activation rate determination can be achieved with static treatment, for which the relative position of the APPJ and the substrate does not change during the plasma-ON time. In this study, the static treatment is mainly discussed. As for activation area, the values determined with the AIR method (see Section 2.2) are taken.

The activation rate calculated according to Equation (1) as a function of the treatment time is shown in the double-logarithmic coordinate system in Figure 8. Both curves, for CDA and for nitrogen, show a drastic increase in the activation rate with decreasing treatment time. The maximum static activation rates measured for 0.5 s treatment time are 4.0 cm$^2$ s$^{-1}$ and 18.2 cm$^2$ s$^{-1}$ for CDA and N$_2$ PDD, respectively. For 10 s treatment, these values are 0.46 cm$^2$ s$^{-1}$ and 1.28 cm$^2$ s$^{-1}$, respectively. Furthermore, the ratio of activation rate for nitrogen and CDA is decreases from 4.7 to 2.7 with treatment time growing from 0.5 to 10 s.

The fitting lines shown in the diagram can not be extrapolated to the values below 0.5 s, because for treatment times $0 < t_0 < 0.5$ s, the activation area must vanish and the activation rate reaches zero. After time $t_{max}$ fulfilling the condition $t_0 < t_{max} < 0.5$ s, the maximum activation rate is reached. Its value is unknown yet, because 0.5 s is the minimum treatment time setting. The expected lower $t_{max}$ value, and consequently, the much higher activation rate for nitrogen, is promising for N$_2$-based high-speed processing.

Not only the absolute values of the activation rate but also the ratio of activation rate for nitrogen to activation rate for CDA increases with decreasing treatment time. Using the formulas of the fitting curves from Figure 8, this ratio increases from 2.5 for 10 s up to 4.0 for 1 s.

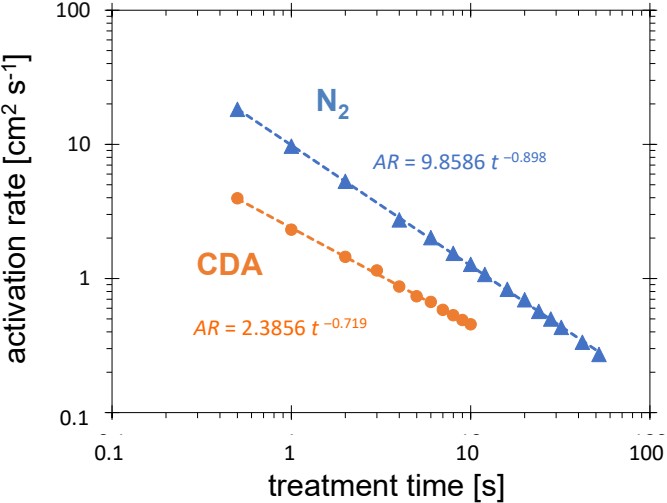

**Figure 8.** Influence of treatment time on the activation rate for CDA and N$_2$ flow of 8.0 SLM, distance between the PCPG tip and the HDPE substrate of 5.5 mm, and the PCPG input power of 8.0 W.

*3.3. Hydrophobic Recovery*

The hydrophobic recovery is understood as the loss of activation due to storge, heat, abrasion or chemical influences. It proceeds at a different rate for each polymer, and the surfaces are still more hydrophilic after many days compared to the original untreated state [72]. Typically, it is defined by the decrease in the SFE during long-term storage. For example, the SFE of 46 mJ/m$^{-2}$ reached on LDPE after corona treatment decreases after 22 days of exposition to air, down to an SFE of 36 mJ/m$^{-2}$ [73,74], in comparison with the SFE for nontreated LDPE which is 31 mJ/m$^{-2}$.

Another way of characterizing the hydrophobic recovery is by the increase in the contact angle determined with the droplet test. For example, the low-pressure oxygen–plasma-treated polyethylene surface shows a contact angle of distilled water increasing from 40° immediately after treatment up to about 65° after five days of storage [63]. In another example, the contact angle reaches 42° after treatment in low-pressure air radiofrequency plasma and increases up to 70° within one day of storage [75]. After treatment with microwave low-pressure nitrogen plasma, a contact angle of 30° is reached, which is 10–20° lower than for oxygen plasma. After 40 days of storage, an angle of about 65° is reached.

In this study, the hydrophobic recovery is defined as reduction in the visualized activation area due to storage. Figure 9 shows the dependence of the activation area on storage time for HDPE surface treated for 10 s with PDD operated with 8 W at 8 SLM nitrogen flow.

Despite the hydrophobic recovery documented for HDPE in the literature on the basis of SFE or contact angle, no significant reduction in the activation area can be observed after storage of the activated substrates over an extended period. The results in Figure 9 document the changes over 668 h. During 24 h of storage, only a minor decrease of the activation area is observed, by less than 4%. This result is not contradictory to the previously reported hydrophobic recovery results, because those are related mainly to the decrease in the maximum value of the SFE reached after plasma treatment. The SFE of 58 mN/m used as a threshold for visualization in this work is much lower than the maximum value of SFE obtained after treatment, reaching an SFE of up to 72 mN/m. The contours with 58 mN/m do not have to be affected much by the decline of the maximum SFE and shrinking of the contours with values larger than 58 mN/m. It is also worth mentioning that the PDD-treated substrates are stored in a dark place, which reduces the hydrophobic recovery.

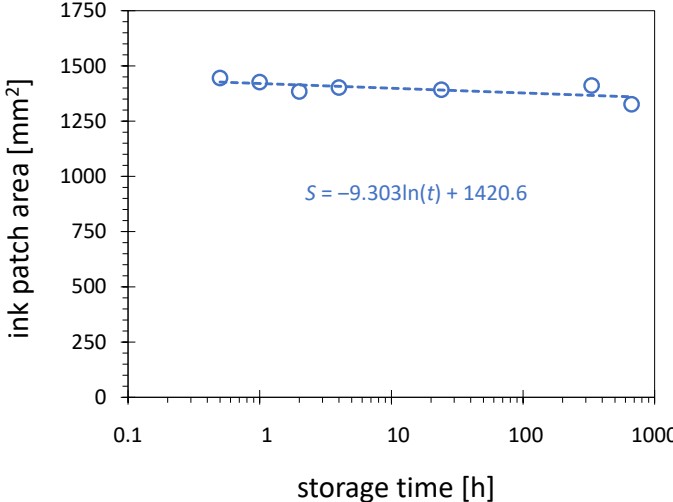

**Figure 9.** Influence of the storage time on the activation area. The 10 s plasma treatment of the HDPE substrates was conducted with nitrogen flow of 8.0 SLM, PCPG input power of 8.0 W, and visualized with the 58 mN/m test ink.

### 3.4. Water Contact Angle

The water contact angle of 101° measured on pristine HDPE used in this study fits well within values cited in literature [76], ranging from 87° to 105°. Any plasma treatment presented in this study, even very minute, results in significant decrease in the contact angle below this value.

### 3.4.1. Influence of Distance

Despite the vanishing of the activation area with 58 mN/m for substrates treated with nitrogen PDD at a distance larger than 20 mm (see Figure 5), some activation with increased SFE (reduced contact angle) is still present, especially for treatment time prolonged up to 2 min. Figure 10 shows the increase in the water contact angle with increasing distance between the PCPG and the HDPE substrate. The minimum contact angle of 41° is reached for the smallest distance of 5 mm. This minimum value is also typical for other types of APPJ operated with nitrogen for HDPE treatment [61].

Between 10 and 30 mm, the contact angle increases up to 81° and stagnates at this level for distances larger than 30 mm. The position of this zone correlates with the drop in the electric field and the limit of the direct discharge. For distances larger than 30 mm, the surface activation is mainly due to the long-living gaseous plasma products. Also, this behaviour is in agreement with results obtained with other types of APPJ operated with nitrogen and used for PE treatment [77].

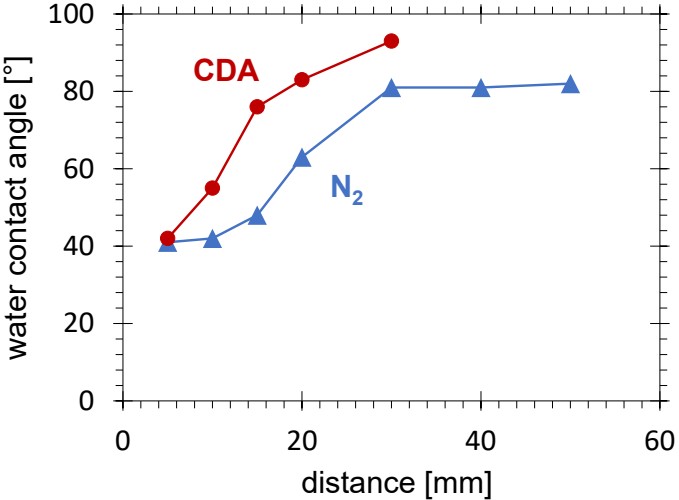

**Figure 10.** Water contact angle measured at the HDPE substrate surface after 2 min treatment with nitrogen and CDA PDD as a function of distance between the PCPG and the substrate, the PCPG input power of 8.0 W, and the gas flow of 10.0 SLM.

The results for CDA are included in Figure 10 for comparison. The contact angle for CDA in the entire investigated distance range is higher than for nitrogen. It increases rapidly much closer to the PCPG, exceeds 80° at 20 mm, and reaches a maximum of 93° at 30 mm. As already discussed in Section 3.1.1, the activation disappears much closer to the PCPG than for nitrogen due to fast quenching of the metastable excited species at high oxygen concentrations.

### 3.4.2. Influence of Treatment Time

The dependence of the contact angle on time was determined for a distance of 15 mm (see the squares in Figure 11). As can be expected [77,78], with increasing treatment time the contact angle decreases. For treatment conditions at which the activation area decreases to one-third of the maximum value (distance: 15 mm, treatment time: 10 s) it is 52°. For very long treatment times of more than 2 min, it converges to 40°.

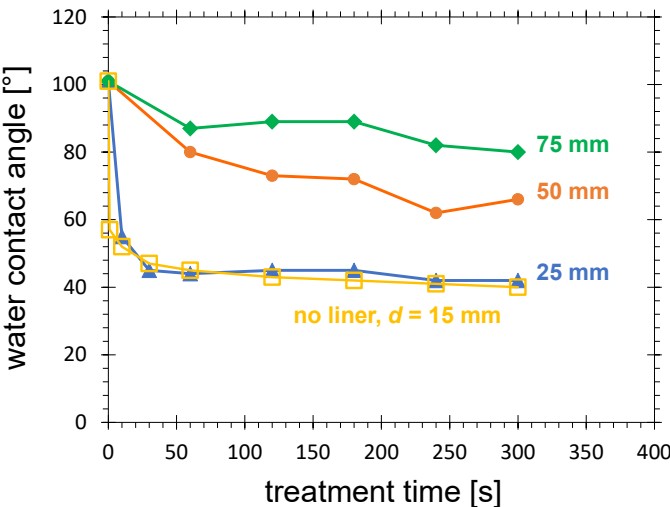

**Figure 11.** Water contact angle measured at the HDPE substrate surface after nitrogen PDD treatment as a function of treatment time for 3 different lengths of the plasma liner extension, and without plasma liner at 15 mm, as specified in the diagram, and the distance between the plasma liner extension and the substrate of 2 mm. Further treatment parameters are the PCPG input power of 8.0 W, and the gas flow of 10.0 SLM.

### 3.4.3. Extended Plasma Liner

The nitrogen APPJ plume can be increased by enclosing the discharge, this way, the mixing of the plasma gas with ambient air is strongly suppressed [79,80]. To increase the lifetime of the metastable excited nitrogen, the tubular extensions of the plasma liner with four lengths, 25, 30, 50 and 75 mm, are used. Only a 2 mm slot is left between the bottom edge of the plasma liner extension (see Figure 1a) and the substrate. Figure 12 shows the plasma plume of the PDD enclosed within the 30 mm-long plasma liner extension for nitrogen flow of 12 and 8 SLM. From these pictures it is apparent that the PDD plasma plume is much larger than for PDD without plasma liner extension. It is also longer for higher gas flow (Figure 12a). The fiberlike microdischarge channels can be recognized. They apparently follow the lines of the electric field, not the gas flow. The fastest gas flow can be expected along the axis of the plasma liner extension, but the microdischarges end at the inner side of the liner extension wall. This surface can be electrically charged by the current flowing through the microdischarge, and hence cause electrostatic attraction for the charged species in the microdischarge. With increasing nitrogen flow, the plasma plume expands. For high nitrogen flow, the concentration of oxygen in the gas decreases and the lifetime of metastable excited species is prolonged. In the environment with a high concentration of the metastable excited species, the electric field needed for sustaining the microdischarges is lower. Consequently, the primary discharge is present in regions more remote from the tip of the PCPG. This observation helps us to understand the influence of the plasma liner extension on contact angle results.

The time-dependent curve for the 25 mm plasma liner extension is almost overlapping with the curve for 15 mm distance without plasma liner extension (see Figure 11). The minimum reachable contact angle increases with length of the plasma liner extension, reaching 62° and 80° for 50 and 75 mm, respectively. For such a large distance, larger than 30 mm, the electric field generated by the PCPG is not strong enough to initiate the microdischarges, even in the gas with a high concentration of metastable excited molecules. Consequently, the location of the generation of the oxidizing species (primary plasma of microdischarges) becomes more remote from the substrate surface, the surface activation becomes weaker, and the contact angle grows.

Apparently, prolonging the treatment time cannot compensate for the loss of treatment efficiency due to the increased distance.

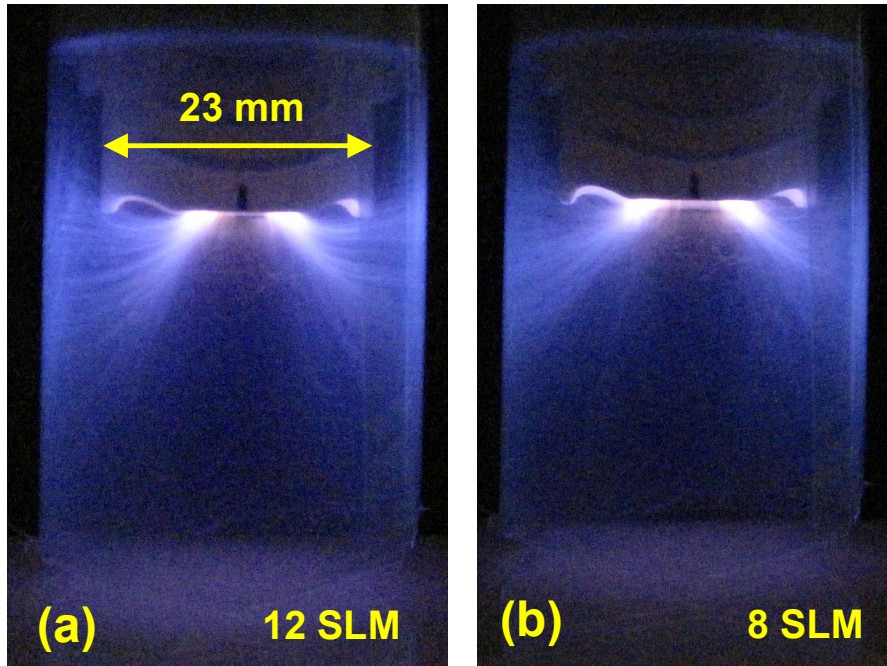

**Figure 12.** The PDD inside the 30 mm-long liner extension for nitrogen flow of (**a**) 12 SLM and (**b**) 8 SLM.

*3.5. PMMA Lenses Treatment*

As an application example, the nitrogen PDD was applied for hydrophilization of a lens panel made of PMMA. Figure 13a shows a nontreated lens panel. It is not wettable with the 58 mN/m formamide test ink. After treatment with piezobrush® PZ3-i operated with nitrogen, the wetting of the entire surface of the panel is achieved (Figure 13b). Due to the optical properties and segmentation of the substrates, the AIR method was not applicable for quantitative evaluation of the plasma treatment effect.

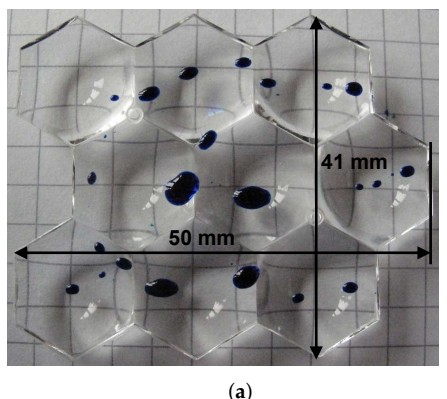

(**a**)

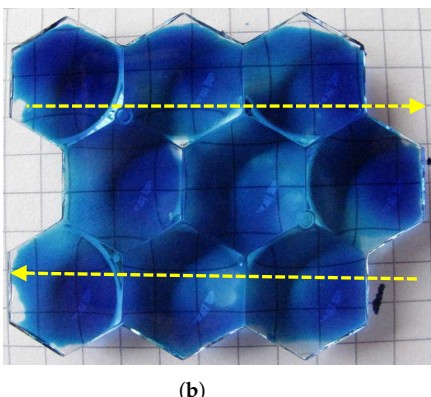

(**b**)

**Figure 13.** The wetting with formamide test ink (58 mN/m) of the PMMA lens panel (**a**) untreated, and (**b**) treated with nitrogen PDD. Treatment conditions: distance 6 mm, nitrogen flow: 8.0 SLM, relative PZ3-i movement as shown by yellow arrows, and movement speed: 20 mm/s.

The alternative method is, again, the contact angle measurement. The untreated PMMA shows an SFE of 41.1 mJ m$^{-2}$ [81], corresponding to the contact angle with distilled water of 76°, as shown in Figure 2a, which is well within range of the PMMA contact angle data summarized in [82], and ranging in temperatures of 20 °C from 60° to 80°.

After 3 s plasma treatment in nitrogen PDD, the contact angle of 37° is reached, as shown in Figure 2b. In [83], a water contact angle on PMMA as low as to 25° after 30 min treatment with Argon APPJ is reported. However, after 3 s of treatment with the

same device, the contact angle remains larger than 50°. Another literature example [84] shows that the contact angle of a pristine PMMA of 74° after 90 s treatment with Ar-based APPJ does not decrease below 44°. This speaks for the strong performance of the PDD with nitrogen.

The entire surface of the $40 \times 50$ mm panel reached an SFE of 58 mN/m after 5 s of treatment (two sweeps of the plasma tool with speed of 20 mm/s), corresponding to the dynamic activation rate of 4 $\text{cm}^2\,\text{s}^{-1}$.

## 4. Conclusions

The surface activation of HDPE using the nitrogen PDD was investigated and compared with air PDD. The static activation area was determined using the AIR method based on visualization of the zone with SFE higher than 58 mN/m. The activation area reached after 10 s treatment by nitrogen PDD was 15 $\text{cm}^2$. A static activation area a factor of 3 higher for nitrogen than for CDA was demonstrated. The proposed explanation of this behaviour is the combination of the wide-reaching strong electric field of the PCPG and the high concentration of the metastable excited molecules $N_2\left(A^3\Sigma_u^+\right)$ promoting ionization by the Pennig effect and extension of the primary discharge.

The static activation rate, calculated as the activation area divided by treatment time, strongly decreases with treatment time. The maximum static activation rates measured for 0.5 s treatment time are 4.0 $\text{cm}^2\,\text{s}^{-1}$ and 18.2 $\text{cm}^2\,\text{s}^{-1}$ for CDA and $N_2$ PDD, respectively. For 10 s treatment, these values are 0.46 $\text{cm}^2\,\text{s}^{-1}$ and 1.28 $\text{cm}^2\,\text{s}^{-1}$, respectively. Also, the ratio of activation rate for nitrogen and CDA decreases from 4.7 to 2.7 with treatment time growing from 0.5 to 10 s.

The hydrophobic recovery on HDPE treated with nitrogen PDD is defined as a reduction in the activation area with storage time. No significant hydrophobic recovery was observed after 668 h of storage in a dark place.

For treatment of substrates at large distance from PCPG, where SFE is below 58 mN/m, the water contact angle is applied for evaluation of the discharge performance. In comparison with other parameters, the contact angle is significantly lower after treatment with nitrogen-based plasma. Application of plasma liner extension allows for increasing the distance from PCPG, with which a reduction in the water contact angle after treatment with nitrogen PDD can be observed. A contact angle of 80° is measured at distances more than 75 mm from the PCPG tip after a treatment time of 5 min. The lens panel made of PMMA is a practical activation example. A dynamic activation rate of 4 $\text{cm}^2\,\text{s}^{-1}$ is demonstrated.

**Author Contributions:** Conceptualization, D.K., F.H. and T.A.; methodology, D.K.; software, T.A.; validation, D.K., F.H. and T.A.; formal analysis, S.G.; investigation, S.G.; resources, F.H.; data curation, S.G.; writing—original draft preparation, D.K.; writing—review and editing, S.G.; visualization, D.K.; supervision, S.L.; project administration, F.H.; funding acquisition, S.L. All authors have read and agreed to the published version of the manuscript.

**Funding:** This research received no external funding.

**Institutional Review Board Statement:** Not applicable.

**Informed Consent Statement:** Not applicable.

**Data Availability Statement:** Data available from the corresponding author on request.

**Acknowledgments:** The PCPGs of CeraPlas™ F type used in this study are provided by TDK Electronics GmbH.

**Conflicts of Interest:** The authors declare no conflict of interest.

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
