# Peer review of "Application of Nitrogen Piezoelectric Direct Discharge for Increase in Surface Free Energy of Polymers"

_plasma, doi:10.3390/plasma5010009_

Round 1

Reviewer 1 Report

This is an interesting paper which provides useful information.

  1. What is the ozone generation situation with this plasma jet? How about the influence of the plasma carrier gas on ozone generation? The environment air may participate in the plasma reaction to generate ozone although no oxygen is in the plasma carrier gas.
  2. What is the substrate temperature during the plasma jet treatments with CDA and N2? Can this also influence the activation area?
  3. 3.1 Definition of activation area. Just to make sure that activation area is defined by the wetting area of the test inks. Is this correct? If so, the authors may need to provide the volume of the testing ink. Will the volume of testing ink influence the experimental results?
  4. Figure 5b. It is interesting that CDA and N2 have quite different trends. From lines 181-188, the authors have explained the competing mechanisms. However, it is still not clear why CDA and N2 have such differences. The environmental air could also participate in the N2 plasma jet reaction to produce ozone. Would the authors please give more detailed explanations.
  5. Figure 5 and Figure 6. It is interesting that N2 plasma jet has larger activation area than CDA one in all experimental vairables. I would expect more ozone generation with CDA. The generated ozone may make the surface more hydrophilic.

Reviewer 2 Report

Dear authors,

The manuscript plasma-1566204, entitled 'Application of nitrogen piezoelectric direct discharge for increase of surface free energy of polymers' presents the PZ3-i plasma source as a direct application for modifying the polymeric surface free energy, for adhesion purposes.  The manuscript is well written, the figures are clear but I suggest tho the authors to make them bigger. The experimental results are very nice and well discussed. The conclusions are in brief and well argued. 

However I expect to see in the results paragraph info on the emission spectra of the discharge, in order to proper link plasma treatment with the effects found on the polymer surface.  Moreover, I suggest the authors to include OES spectra of the discharge when treating polymeric samples. If possible to measure also the O3 and NOx during plasma treatment of the surface. The insights given by the plasma active and reactive species, along with the O3 and NOx could help in better discussion of plasma effects on the free energy modification of the polymer surface.

With these added info the manuscript could be taken into consideration for publication in Plasma journal. 

Minor revision
